# GROUNDING CODE GENERATION WITH INPUT-OUTPUT SPECIFICATIONS

## ABSTRACT

Large language models (LLMs) have demonstrated significant potential in code generation. However, the code generated by these models occasionally deviates from the user's intended outcome, resulting in executable but incorrect code. To mitigate this issue, we propose GIFT4CODE, a novel approach for the instruction fine-tuning of LLMs specifically tailored for code generation. Our method leverages synthetic data produced by the LLM itself and utilizes execution-derived feedback as a key learning signal. This feedback, in the form of program input-output specifications, is provided to the LLM to facilitate fine-tuning. We evaluated our approach on two challenging data science benchmarks, ARCADE and DS-1000. Our results suggest that the method enhances the LLM's alignment with user intentions, considerably reducing the incidence of executable but incorrect outputs. Consequently, this leads to a marked improvement in the quality of generated code.

## 1 INTRODUCTION

Large language models (LLMs) trained on code have demonstrated tremendous success as AI pair programmers in assisting developers writing code (Chen et al., 2021a; Austin et al., 2021; Li et al., 2023; Chowdhery et al., 2022; Li et al., 2022; Nijkamp et al., 2022; Fried et al., 2022; Li et al., 2023). Developers often interact with code LLMs using succinct natural language (NL) intents (*e.g.* $x$ in Fig. 1) to describe their tasks (Barke et al., 2022; Ross et al., 2023). However, NL intents are often ambiguous (Yin et al., 2022b). This ambiguity can be problematic in complex tasks, such as manipulating Pandas DataFrames or PyTorch Tensors (Lai et al., 2022).

Auxiliary input-output (I/O) specifications, ranging from concrete I/O examples to high-level NL summaries (*e.g.* red text in Fig. 1), offer a natural way to reduce this ambiguity (Gulwani et al., 2015; Balog et al., 2016; Jain et al., 2022; Yin et al., 2022a). Prior to the emergence of LLMs, auxiliary specifications served as essential problem descriptions in program synthesis (Gulwani, 2016; Devlin et al., 2017; Shi et al., 2020). Real-world synthesis systems like FlashFill are testimony to the adoption and effectiveness of I/O specifications (Gulwani, 2011; Gulwani et al., 2012). In this work, we consider the problem of LLM-based code generation when the LLM has access to both a natural-language intent and an additional I/O specification.

However, code LLMs often fall short on following intents with additional complex semantic constraints like I/O specifications out-of-the-box, leading to plausible solutions that fail to satisfy the constraints (*e.g.* $y'$, Fig. 1). Such a lack of *alignment* between the user's intent and the model's predictions (Chen et al., 2021a) could pose unnecessary burden on developers who are then required to fix the generated code (Bird et al., 2023). Therefore, we posit that addressing this misalignment by *grounding* the code generated by LLMs to the provided specifications is of paramount importance.

Instruction fine-tuning has emerged as an effective strategy to tackle the issue of misalignment (Wei et al., 2021; Sanh et al., 2021; Chung et al., 2022). Classical approaches for instruction tuning typically require a substantial amount of parallel labeled data of NL intents and gold model responses. The process of gathering such data is labor-intensive and time-consuming. Recent studies have suggested that generating synthetic instruction-following data using the LLM itself is a promising approach to improve alignment, with empirical success on natural language text generation tasks (Wang et al., 2022a; Honovich et al., 2022a; Taori et al., 2023; Peng et al., 2023, *inter alia*).

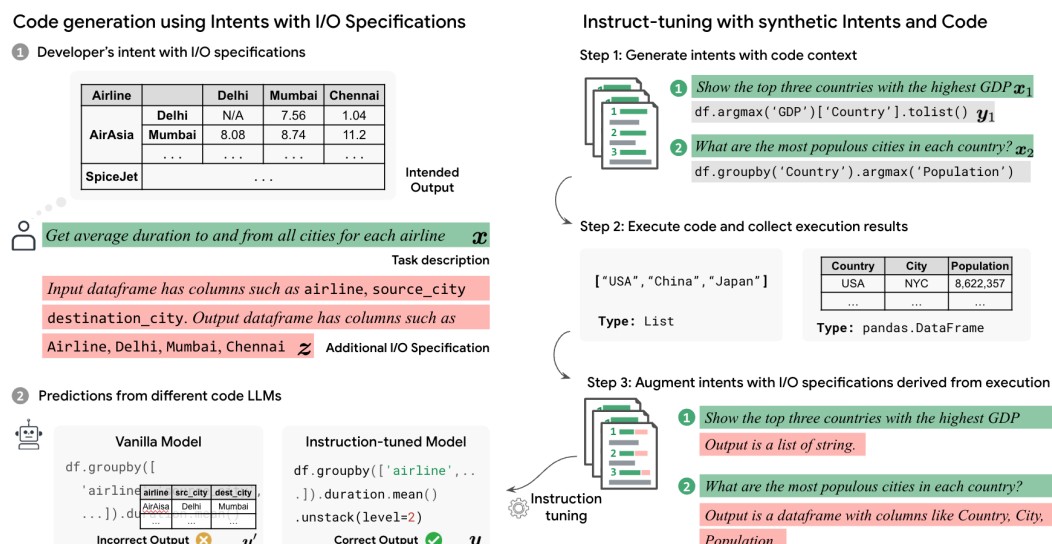

Figure 1: **Left**: Illustration of how developers prompt code LLMs with NL intents and I/O specifications to generate code with complex outputs (`pandas Dataframes`). Vanilla code LLMs fail to understand extra I/O specifications. **Right**: Our proposed instruction tuning approach uses synthetic intents and code solutions, where intents are augmented with I/O specifications derived from program execution results. Models trained on the synthetic data could better follow a developer's intent.

In this paper we build upon the recent success of instruction tuning using synthetic data and fine-tune code LLMs to follow NL intents with additional I/O specifications. Unlike existing approaches, our key insight is to leverage program execution for synthetic data generation. First, in contrast to other open-ended text generation tasks where assessing the quality of target responses is challenging, the quality of synthetic code generation data can be easily improved using heuristics such as code executability (Yin et al., 2022c). Moreover, from the program execution states one could derive precise and *aligned* I/O specifications that can be included in the intents to supervise a model to follow those extra semantic constraints (Fig. 1, *Right*). In other words, when fine-tuned on such synthetic data, a model learns to *ground* NL task descriptions to program execution states expressed as I/O specifications (Berant et al., 2013).

We apply our grounded instruction fine-tuning for code (GIFT4CODE) method to two challenging natural language to code generation applications: synthesizing complex `pandas` programs in computational notebooks (ARCADE, Yin et al. (2022b)) and answering data science questions on Stack Overflow (DS-1000, Lai et al. (2022)). First, we demonstrate the value of leveraging program execution information by showing that strong code LLMs can already be significantly improved by up to 10% absolute on ARCADE after fine-tuning on intents and executability-filtered code solutions *without* including any I/O specifications in synthetic data. Then, to further align model predictions to various types of user-provided I/O specifications, we derive those specifications at different levels of abstraction from code execution results. This ranges from concrete input/output examples to succinct natural language summaries of target variables (specifications in Fig. 1). By fine-tuning on parallel data of intents with I/O constraints and their target code solutions, the model is better at following a developer's intents while producing code that is more likely to execute to the desired outcome.

## 2 PROBLEM FORMULATION

**Natural Language to Code Generation**  Code generation considers the task of translating a developer's natural language intent $x$ into a machine-executable program $y$ (*e.g.* Fig. 1, *Left*). An intent usually contains a succinct and potentially ambiguous task description. For tasks with complex outputs, the intent may also include additional I/O specifications as extra clarifications.[1] Code generation tasks are often contextualized, meaning that an intent is associated with certain *programmatic*

---

[1]For simplicity, we consider I/O specifications as part of the intent hereafter.

*contexts* $c$ (Iyer et al., 2018), such as the code that a developer has already written in an IDE prior to the intent (*e.g.* df = pd.read_csv("flights.csv") for $x$, not shown in Fig. 1). Intuitively, a model needs to leverage both the intent and the programmatic context (*e.g.* variable df) to generate a suitable code solution.

**Supervised Instruction Tuning for Code LLMs**   Supervised instruction tuning aims to improve code LLMs by fine-tuning them on parallel data of intents and target code solutions. In this paper we consider automatically synthesizing such parallel data by prompting LLMs using few-shot demonstrations (other approaches are discussed in §5). Specifically, the synthetic dataset consists of examples $\{\langle c, x, y \rangle\}$ of intents $x$ with programmatic contexts $c$ and their generated code solutions $y$.

## 3   GIFT4CODE: LEARNING TO FOLLOW INTENTS WITH I/O SPECIFICATIONS

In this section we elaborate on GIFT4CODE, our proposed approach to fine-tune code LLMs to better follow developers' natural language intents along with I/O specifications, using synthetic parallel data. Fig. 1(*Right*) illustrates an overview of GIFT4CODE. We first synthesize a collection of intents with code solutions via few-shot prompting (§3.1), and then execute model-predicted code to derive I/O specifications from execution results (§3.2). Finally, we fine-tune the code LLM to predict code solutions given intents inlined with I/O specifications (§3.3).

### 3.1   GENERATING SYNTHETIC INTENTS AND CODE SOLUTIONS

**Programmatic Contexts**   We initialize a program state given some programmatic context and generate a series of contextualized NL-to-code problems for that context.  As an example, the synthetic problems in Fig. 1 (*Right*) could have the contextual code df = pd.read_csv("world_statistics.csv"), which initializes the DataFrame variable df, subsequently used in the generated synthetic examples. The fact that our problems are contextualized sets our approach apart from existing instruction-tuning methods for text generation models (Wang et al., 2022a; Honovich et al., 2022a), where synthetic examples do not depend on any particular contexts. In our case, we mine those programmatic contexts from real-world code repositories, such as tabular datasets (e.g., .csv) used in data science notebooks on Github (§4).

**Creating Initial NL Intents**   Given a programmatic context $c$, we few-shot prompt an LLM to create a sequence of natural language intents $\{x_i\}$ (*e.g.* $x_1$, $x_2$ in Fig. 1(*Right*)). A problem $x_i$ that appears later in the sequence might depend on the earlier ones $\{x_{<i}\}$ (Nijkamp et al., 2022; Yin et al., 2022b). To generate NL intents, we use a "generalist" LLM instead of the code LLM that we aim to improve, since predicting intents conditioned on some context is similar to other text generation tasks, which could be better handled by a LM trained on general-purpose text data (Zelikman et al., 2022). The "generalist" LLM is a state-of-the-art general-purpose large language model. It achieves competitive results with GPT-4 on a variety of NL reasoning tasks. [2] Empirically, we observe that the problems generated by this LLM encompass a wide range of tasks relevant to the given programmatic context. Readers can refer to Appendix B for examples. We remark that those model-predicted intents do not come with I/O specifications yet.

**Predicting Code Solutions**   After generating an intent $x$, we then prompt the code LLM to get a code solution $y$ for $x$ (*e.g.* $y_1$ in Fig. 1(*Right*)). Specifically, a prompt to the LLM is the concatenation of the programmatic context $c$ and the intent $x$, with additional few-shot demonstrations of $\{\langle c', x', y' \rangle\}$. Since many NL intents can be ambiguous and there could exist multiple alternative solutions (*e.g.* without additional I/O specifications, the intent in green in Fig. 1(*Left*) could be answered using tables with different layouts; see more in Yin et al. (2022b)), we therefore draw multiple candidate code solutions $\{y\}$ for each intent. Intuitively, $\{y\}$ could have a variety of alternative solutions for $x$, each leading to different execution results. This equips the model with the capacity to predict code for the same task but with different user-provided I/O specifications.

---

[2]Details are publicly available in Anonymous (2023b). The model is now publicly available as an API, but was only privately accessible at the time of submission. Anonymized for double-blind review.

| Spec. Type | Description | Example I/O Specification |
|---|---|---|
| TypeDesc | Variable type name | Generate a variable with name `df` and type `pandas.DataFrame` |
| I/O Examples | Concrete I/O examples | Output variable `df`:

`\|Bangalore(float)\|Chennai(float)\|Delhi(float)\|Hyderabad`
`(float)\|Kolkata(float)\|Hyderabad(float)\|Kolkata(float)\|...`
`\|-----\|-----\|-----\|-----\|-----\|-----\|`
`\| nan \| 1.04 \| 8.08 \| 3.62 \| 7.56 \| 7.56 \| 8.32 \|`
`\| 1.18 \| nan \| 11.96 \| 6.80 \| 6.31 \| 8.75 \|`
`\| 8.46 \| 11.10 \| nan \| 9.19 \| 9.52 \| 10.32 \|` |
| I/O Summary | LLM-generated NL summaries of I/O examples | Given the user intent and the code, the salient columns (at most given 3) in the input dataframe are airline, source_city, destination_city. The output dataframe has columns (at most given 3) such as Delhi, Mumbai, Chennai. |

Table 1: Types of I/O specifications proposed in this work at different levels of abstraction. Example specifications are for the intent in Fig. 1(*Left*). Only the output specifications for I/O Examples are shown for brevity.

**Improving Quality of Synthetic Data** The quality of synthetic data is a fundamental issue of data augmentation in instruction tuning (Wang et al., 2022a; Honovich et al., 2022a), and existing approaches in text generation typically resort to simple and noisy heuristics (*e.g.* rejecting examples with different inputs but the same output). As motivated in §1, for NL-to-code generation, we can reliably enhance the quality of candidate code solutions by leveraging inherent program properties, such as filtering out any code that is not executable given the provided programmatic context.

## 3.2 CODE EXECUTION AND INFERENCE OF I/O SPECIFICATIONS

Given the set of synthetic problems $\{\langle \boldsymbol{x}, \{\boldsymbol{y}\}\rangle\}$ generated by few-shot prompting, we execute the code for each problem (step 2, Fig. 1(*Right*)) and derive I/O specifications from the execution results as additional semantic constraints to be included in the intents (step 3, Fig. 1(*Right*)).

Specifically, for each candidate solution $\boldsymbol{y}$ of an intent, we first execute its original programmatic context $\boldsymbol{c}$, followed by executing $\boldsymbol{y}$. We trace the execution to collect the set of input and output variables in $\boldsymbol{y}$, denoted as $\{v\}$, which are used to derive I/O specifications (details below). Executing code with arbitrary programmatic contexts collected from the wild is highly non-trivial due to issues such as library dependency. However, the use of synthetic data alleviates the need for a complex environment setup.

Given the set of input and output variables extracted from execution results, we formulate an I/O specification, denoted as $\boldsymbol{z}$, which serves as additional information to augment a developer's intent, thereby providing a more comprehensive problem description. The level of detail and the style of these I/O specifications can vary based on the complexity of the problem and the developer's preferences. In this work, we investigate three distinct types of I/O specifications, each characterized by its own linguistic style and level of abstraction, as illustrated in Tab. 1.

First, as a simple baseline, we utilize the variable type (TypeDesc, Tab. 1) as the I/O specification. Next, we incorporate the concrete values of the input/output variables into the specification, which we refer to as I/O Examples. This is reminiscent of classical program synthesis using I/O examples (Gulwani et al., 2012; Alur et al., 2013; Balog et al., 2016). However, in our scenario, these I/O examples are used in conjunction with natural language (NL) intents to define the problem, in line with Jain et al. (2022). Given that the majority of the problems in our synthetic dataset involve complex Python objects such as `pandas DataFrames`, we simplify the I/O specification to include only partial variable states (*e.g.* by excluding some rows and columns in large `DataFrames`). Please refer to Appendix C for further details.

In our effort to generate a more natural variety of I/O specifications that closely resemble the style of specifications in developers' NL intents, we employ an LLM to summarize the values of input/output variables $\{v\}$ into a succinct natural language description $\boldsymbol{z}$ (I/O Summary). Intuitively, the NL I/O summary includes salient information in the variables that can best clarify the original intent (*e.g.* the subset of columns in a `DataFrame` that are most relevant to solve a problem, as in Tab. 1, *Bottom*).

Specifically, we few-shot prompt the generalist LLM to generate $z$, using information from its programmatic context $c$, the intent $x$, the code solution $y$, as well as I/O variables $\{v\}$, *i.e.* $z \sim P_{\text{LLM}}(\cdot \mid c, x, y, \{v\})$. We then update the intent $x$ by appending $z$ to it. The few-shot exemplars used for prompting cover example I/O summaries for various types of Python objects, such as nested container types (*e.g.* nested `dicts`), along with more complex objects like `pandas DataFrames` and `pytorch` or `tensorflow Tensors`. See Appendix C for additional details.

### 3.3 Fine-tuning Code LLMs to Follow Intents with I/O Specifications

Our approach, GIFT4CODE, aims to fine-tune code LLMs to generate code that adheres closely to the desired intents which are supplemented by I/O specifications. In our synthetic training data, each example $\langle c, x, y \rangle$ consists of a programmatic context $c$, an intent $x$ augmented with I/O specifications, and the corresponding code solution $y$. During fine-tuning, the code LLM learns to generate code that not only satisfies the provided intents but also respects the specified I/O constraints, while leveraging any relevant information in the programmatic contexts. In other words, we optimize $P_{\text{LLM}}(y \mid c, x)$. It is worth noting that the code LLM that undergoes this optimization is different from the "generalist" LLM employed to generate the NL intents and I/O specification $z$.

## 4 Experiments

The core research question explored in this section is whether GIFT4CODE enhances the LLM's ability to follow developers' NL intents with **complex** I/O specifications. While common code generation benchmarks like HumanEval and MBPP Chen et al. (2021a); Austin et al. (2021) feature simple algorithmic tasks (e.g., sorting) utilizing basic Python data types (e.g., lists), thus allowing for the use of concrete I/O examples as specifications, they lack the diverse and complex I/O specifications that we aim to explore. For more open-ended tasks such as data science programming, the output data type is more complex and diverse (e.g., Pandas DataFrames, PyTorch tensors). Hence, we apply our method to two different data science code generation applications.

**ARCADE** (Yin et al., 2022b) is a benchmark of natural language to code generation in interactive data science notebooks. Each evaluation notebook consists of a series of interrelated NL-to-code problems in data wrangling (*e.g. "Min-max normalize numeric columns"*) and exploratory data analysis (*e.g.* intents in Fig. 1) using the `pandas` library. ARCADE features succinct NL intents to reflect the style of ephemeral queries from developers when prompting LLMs for code completion. More than 50% of the dataset's problems are under-specified, which means that additional I/O specifications could provide extra clarification. To construct programmatic contexts for synthetic training data generation, we scraped 7,500 CSV files that are used in public Jupyter notebooks. Each context contains a DataFrame import statement, for example, `df = pd.read_csv(·)`, followed by an NL description of the DataFrame to help the LLM understand its content. We generated 6 intents for each programmatic context and sampled 5 candidate code solutions for each intent. Roughly 60% of the code samples were executable. After filtering based on executability and API diversity (§3.1), we obtained around $20K$ synthetic training examples.

Our synthetic data only comprises pairs of questions and code samples which lack rich context. To avoid regression in context understanding during instruction fine-tuning, we crafted a mixture dataset which combines the synthetic data and the Python data used to fine-tune the code LLM. Note that this Python data does not contain any execution-based signals or I/O specifications. After approximately 1,500 instruction tuning steps with a batch size of 64, the model reaches its optimal performance. This process consumed about 1.5 epochs of our synthetic dataset.

**DS-1000** (Lai et al., 2022) is a benchmark of data science problems sourced from Stack Overflow (SO). Compared to ARCADE, problems in DS-1000 feature a wider variety of I/O types, such as `numpy`/`scipy Arrays` and `pytorch`/`tensorflow Tensors`, making it particularly appealing to evaluate our instruction tuning approach aimed at generating code following I/O specifications. However, in contrast to ARCADE which features succinct NL intents, DS-1000 follows the typical style of detailed problem descriptions found in SO posts. These elaborate descriptions often include additional information such as task background and descriptions of unsuccessful attempts, providing a more complex intent structure, with an average length of 140 words. Given that such elaborate intents may not reflect the style of developers' prompts to code LLMs, we do not focus on generating

intents with similar styles. Instead, we held-out 500 problems in DS-1000 and use their annotated intents as training data, while evaluating on the remaining problems.[3]

## 4.1 SETUP

**Base Code LLM**    We use a strong decoder-only code language model with 62B parameters. The model was first pre-trained on a collection of 1.3T tokens of web documents and github code data, and was then fine-tuned on a disjoint set of 64B Python code tokens together with 10B tokens from Python Jupyter notebooks (Anonymous, 2023a).[4]

**Learning Methods**    We evaluated the performance of both the baseline and instruction-tuned models across a range of data formats, as shown in Tab. 2. For each I/O specification type, we augmented the intents and few-shot exemplars with specifications of the corresponding type. Similarly, at test time, we augmented the intents with the same type of I/O specifications. The baseline models are tested under both zero-shot and few-shot prompting. For the latter, we manually created exemplars for all types of specifications. These exemplars were prepended to the prompt when querying the LLM for code generation during inference.

**Simulate Noisy I/O Specifications at Test Time**    At testing time, the generation of I/O Summary underwent a minor modification from the process detailed in §3.2. We remove the concrete input/output variable states $\{v\}$ to produce noisy I/O summaries, simulating scenarios where users might give noisy I/O specifications (Devlin et al., 2017). We illustrate an example in Appendix C where the LLM generates an imperfect specification. While the "generalist" LLM uses the code solution to generate noisy I/O summaries, we remark that the code LLM, which we aim to evaluate, does not have access to the ground truth solution. In other words, the "generalist" LLM acts merely as a "data labeler" to create I/O summaries in prompts in order to construct the evaluation dataset. It is also a common practice in program synthesis to derive specifications from ground truth solutions, which then serve as the sole input to the model during its evaluation (Balog et al., 2016).

**Metrics**    We adopted the *pass@k* metrics as defined in Chen et al. (2021a); Austin et al. (2021), which is calculated as the fraction of problems with at least one correct sample given $k$ samples. Following Yin et al. (2022a), we drew 50 samples to calculate *pass@5* and *pass@20* to reduce the variance in ARCADE. Similar to Lai et al. (2022), we drew 40 samples to calculate *pass@1* on DS-1000. Consistent with the original works' settings, the sampling temperature was set to 0.8 for ARCADE and to 0.2 for DS-1000 respectively.

## 4.2 MAIN RESULTS

Tab. 2 presents the *pass@k* results on ARCADE and DS-1000. We evaluate both few-shot prompting and fine-tuning with synthetic data. Specifically, for ARCADE we evaluate on two versions of the dataset. First, we consider the original version where an intent is prefixed by prior notebook cells as its programmatic context (**Full Context**), as well as a **No Context** ablation to simulate the scenario where users query a code LLM using an intent without any context. This no-context setting is more challenging, where the zero-shot performance of the base code LLM is nearly halved. The standard errors in all cells of the table are less than 0.5%, and are excluded for clarity in presentation.

In our few-shot prompting experiments, we observe that *pass@k* generally improves with more detailed I/O specifications. Interestingly, on ARCADE, the improvements from prompting using I/O specifications compared to the baseline where no I/O specifications were used (**no spec**), are more notable in the more challenging no-context scenario (*e.g.* $15.96 \mapsto 23.75$ *v.s.* $30.98 \mapsto 37.11$ for +I/O Examples). This trend suggests that additional specifications could provide more valuable clarifications when adequate programmatic contexts are lacking.

Next, we fine-tune the base code LLM using our synthetic parallel data using different types of I/O specifications. Interestingly, without using any I/O specifications in the synthetic intents, on ARCADE the model already registers significant improvements compared to the zero- and few-shot settings.

---

[3]We only use the annotated intents, while the code solutions and I/O specifications are still predicted by the LLM. We ensure the training and evaluation problems are disjoint and from different SO posts.

[4]Model details are available in Anonymous (2023a), but withheld from this submission for review.

| Methods | ARCADE | | | | DS-1000 |
| | pass@5 | | pass@20 | | pass@1 |
| | No Context | Full Context | No Context | Full Context | |
|---|---|---|---|---|---|
| *Zero-shot Prompting* | | | | | |
| Code LLM (no spec.) | 12.45 | 24.67 | 19.85 | 37.47 | 22.62 |
| *Few-shot Prompting* | | | | | |
| Code LLM (no spec.) | 15.96 | 30.98 | 26.35 | 42.30 | 23.92 |
| + TypeDesc | 16.58 | 29.68 | 29.68 | 42.30 | 25.90 |
| + I/O Examples | 19.85 | 32.47 | 30.79 | 43.23 | **26.41** |
| + I/O Summary | **23.75** | **37.11** | **34.50** | **46.75** | 26.25 |
| *Synthetic Data Fine-tuning* | | | | | |
| Code LLM (no spec.) | 20.78 | 34.33 | 33.40 | 46.94 | 24.56 |
| + TypeDesc | 21.52 | 36.73 | 33.58 | 48.61 | 27.35 |
| + I/O Examples | 25.23 | 42.30 | 38.03 | 53.99 | 28.66 |
| + I/O Summary | **28.01** | **43.79** | **43.04** | **55.47** | **29.34** |
| StarCoder 15B | 11.75 | 22.38 | 17.24 | 32.52 | 26.52 |
| WizardCoder 15B | 12.45 | 24.04 | 18.58 | 34.30 | 27.35 |

Table 2: *pass@k* on ARCADE and DS-1000. For each type of I/O specification in Tab. 1 (*e.g.* +I/O Summary), intents are augmented with I/O specifications of that type (*e.g.* intents inline with I/O summary) in fine-tuning data or few-shot exemplars. At test time, input intents use the same type of I/O specifications.

The model-predicted code solutions are filtered using executability heuristics, which helps improve the quality of the synthetic data, and a model fine-tuned on such data could generally be better at following users' intents, even without I/O specifications. Moreover, by fine-tuning the model to follow intents with additional I/O specifications, we observe significantly better results. We also remark that instruction fine-tuning using natural language I/O summaries (+I/O Summary) yields the best results on both datasets. Intuitively, those I/O summaries could encode salient information in target input and output variables through natural language descriptions, which could make it easier for the model to capture patterns in the data as compared to other more elaborate versions such as using concrete I/O examples.

We also evaluated Starcoder (Li et al., 2023) and its instruction tuned variant WizardCoder (Luo et al., 2023) on ARCADE and DS-1000. The result shows that GIFT4CODE is a more effective instruction tuning method in the data science domain. This is especially observed by the fact that GIFT4CODE offers much more relative improvement to the base model than the gains WizardCoder boasts over StarCoder. Overall, our results demonstrate that GIFT4CODE significantly improves the performance of code LLMs in following intents with I/O specifications at varying level of abstraction.

### 4.3 QUALITATIVE ANALYSIS

To gain deeper insights into the behavior of the different models, we present a qualitative analysis of the baseline model, its few-shot prompting variant with LLM specifications (I/O Summary), and our proposed GIFT4CODE approach. We illustrate this analysis using two examples from the ARCADE dataset, as shown in Fig. 2.

In the first example (Fig. 2, *Left*), the base code LLM (cell 4) fails to group the "League" column as requested in the user's intent. Note that the provided solution is still executable so it cannot be filtered by executability heuristics. The few-shot prompting variant with I/O summary (cell 5) also fails here. It struggles to correctly utilize these specifications despite selecting the correct salient columns, leading to an output that does not meet the user's requirement either. In contrast, the output from GIFT4CODE (cell 6) successfully generates a solution which computes the sum of the two salient columns then sorts the result, effectively utilizing the provided specifications and adhering to the user's intent.

The second example (Fig. 2, *Right*) further underscores the advantages of GIFT4CODE. The baseline code LLM (cell 4) attempts to call a non-existing column (`Turbo_Type`), leading to a `KeyError`. This represents a failure mode that the model tries to adhere a user's intent but generates an in-executable solution that refers to incorrect input variables due to lack of I/O specifications. The few-shot prompting variant (cell 5) presents another interesting failure mode. While the model is

```
[1] import pandas as pd
    import numpy as np

    df=pd.read_csv('dataset/full_data.csv')
    df=drop(['Unnamed: 0'],axis=1)

[2] # Schema of Dataframes:
    # Columns in df with example values:
    # name (Mike), subject (math), score (90), date (05-01),
    Time (16:00), H_Score (4.0), A_Score (1.0)

[3] What is the maximum total score in each league?

    # LLM Generated Spec: The output is pandas.core.series.Series
    of type float64. Given the user intent and the code, the
    salient columns (at most given 3) in the input dataframe are
    League, H_Score, A_Score.

[4] Code LLM (irrelevant code)

    df[df['League']=="premier-league"].groupby(
    'Date').sum()[['H_Score','A_Score']].sort_values(
        by=['H_Score','A_Score'],ascending=False)

[5] I/O Summary Few-shot (sorts the wrong column)

    df.groupby('League')[['H_Score','A_Score']].sum().sort_v
    alues('H_Score',ascending=False).head(1)

[6] GIFT4Code (sums and sorts the correct columns)

    df['Total_Score'] = df['H_Score'] + df['A_Score']
    df.groupby('League')['Total_Score'].max().sort_values(

        ascending=False)
```

```
[1] import pandas as pd
    df=pd.read_csv('dataset/car_price_prediction.csv')
    df['Mileage'] = df['Mileage'].apply(
        lambda x : str(x).replace("km"," ")).astype(int)

[2] # Schema of Dataframes:
    # Columns in df with example values: Manufacturer (LEXUS),
    Model (RX 450), Category (Jeep), Engine volume (3.5)

[3] How many turbo jeep cars does Lexus have?

    # LLM Generated Spec: Output is an int. Given the user intent
    and the code, the salient columns (at most given 3) in the
    input dataframe are Engine volume, Category, Manufacturer.

[4] Code LLM (KeyError, calling a non-existing column)

    df[(df['Manufacturer'] == 'Lexus') & (
        df['Model'] == 'RX 450') & (df['Turbo_Type'] ==
    'Turbo') & (df['Category'] == 'Jeep')]

[5] I/O Summary Few-shot (syntax error (in red))

    df[(df.Manufacturer == 'Lexus') & (df.Category ==
    'Jeep') & (df.Engine volume.str.contains(
        'turbo', case=False))].shape[0]

[6] GIFT4Code (correctly uses the specs)

    df[(df['Category'] == 'Jeep') & (
        df['Manufacturer'] == 'LEXUS') & (
            df['Engine volume'].str.contains('Turbo'))].shape[0]
```

Figure 2: Two examples on ARCADE. **Left**: the base code LLM does not follow the intent to group the "League" column. Its few-shot prompting variant failed to correctly utilize the specifications. GIFT4CODE's solution aligns to the user's intent. **Right**: code LLM tries to call a non-exsiting column, leading to a `KeyError`. Its few-shot variant follows the specifications incorrectly, leading to the syntax error. GIFT4CODE generates correct solution.

trying to follow the additional I/O specification (presumably because of the few-shot demonstrations) by referring to the `Engine volume` column in the specification, it fails to generate a syntactically correct code snippet (`df.Engine volume`). It is important to note that this is a common failure mode of the few-shot prompting model, as we explain in Fig. 3 later. Once again, GIFT4CODE outperforms other settings, generating solutions that answer the natural language question while following the additional specifications.

## 4.4 EXECUTION RATE VS $Pass@k$

We delve further into the experimental results to examine the relationship between executability and the quality of the generated code. Surprisingly, we observe that a model with a higher execution rate does not necessarily produce better code. Fig. 3 plots the frequency of common error types alongside the code execution rates of different models' predictions. The baseline code LLM, despite generating a substantial amount of executable code (higher ★), often produces incorrect (irrelevant) solutions, leading to a high executable rate but low $pass@k$ accuracies (Tab. 2). This suggests that a model's ability to generate executable code does not necessarily indicate its competence in generating semantically correct code that aligns with the user's intent. This insight is further evidenced when we fine-tune the model on

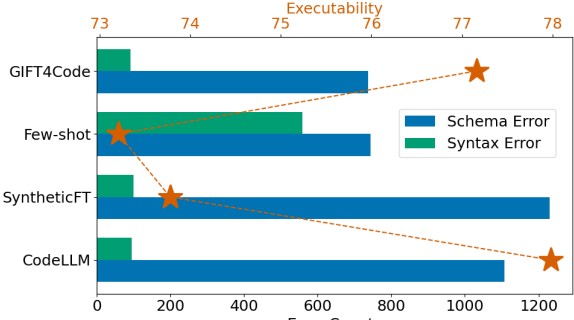

Figure 3: Frequency of error types and code execution rate for different methods. Bottom x-axis stands for the counts of schema errors and syntax errors. Top x-axis represents execution rate. Instruction fine-tuning without specifications (SyntheticFT) decreases executability. Few-shot prompting with specifications reduces schema understanding errors with more syntax errors. GIFT4CODE achieves the best performance by combining their strengths.

synthetic data without I/O specifications, labeled as **SyntheticFT** in Fig. 3. The model's execution rate decreases in this scenario because it attempts to better align with user intent, leading to a higher rate of schema understanding errors (*e.g.* referring to non-existing columns as in cell 4, Example 2,

Fig. 2). However, the *pass@k* scores of this synthetic fine-tuned model remain significantly higher than the base code LLM in Tab. 2, demonstrating the importance of intent alignment.

Incorporating I/O specifications using few-shot prompting leads to another interesting observation. We observe a reduction in schema understanding errors, indicating that the model indeed benefits from the extra specifications. However, given the vast diversity of Python I/O specifications, it is impossible to cover all variants within the few-shot exemplars. Consequently, the model struggles to reliably leverage the specifications, leading to a surge in syntax errors when referring to arbitrarily string-valued schema elements (*e.g.* cell 5, Example 2, Fig. 2). GIFT4CODE effectively mitigates these syntax errors, leading to a higher execution rate while achieving the best *pass@k* accuracies.

## 5    RELATED WORK

**Execution Guided Code Generation**    One area of study primarily focuses on utilizing execution as I/O examples, facilitating the synthesis of programs that align with the intended behavior. Gulwani (2016) involves synthesizing intended programs in an underlying domain-specific language (DSL) from example based specifications. This method has been further explored and adapted to different applications in subsequent studies (Devlin et al., 2017; Chen et al., 2018; Bunel et al., 2018). Another strand of research (Chen et al., 2021b; Wang et al., 2018; Ellis et al., 2019) leverages intermediate execution results to guide the search of programs. More recently, there have been attempts to utilize program execution results to verify and select code samples predicted by LLMs, either during auto-regressive decoding to prune search space (Zhang et al., 2023), or by few-shot prompting (Chen et al., 2023) and post-hoc reranking (Shi et al., 2022; Ni et al., 2023).

**Instruction Fine-tuning**    Instruction fine-tuning is a widely adopted approach to address the mis-alignment issue in LLM-generated content. LLMs such as FLAN (Wei et al., 2021), which excel at understanding and executing instructions from prompts, are trained on labeled training data. Re-inforcement learning with human feedback (RLHF) aims to mitigate the amount of labeling effort using model-based reward (Ouyang et al., 2022). Other works also confirmed the effectiveness of using instructional data in the fine-tuning stage (Mishra et al., 2021; Sanh et al., 2021; Chung et al., 2022; Wang et al., 2022b). To lower labeling cost, several recent works explored the possibility of automatic instruction generation (Ye et al., 2022; Zhou et al., 2022; Honovich et al., 2022b). In particular, SELF-INSTRUCT (Wang et al., 2022a) demonstrated that LLMs can be further improved by utilizing its own generation of instruction data. Our work differs from this line by considering execution-based specifications. Additionally, recent works attempted to distill instruction following data from more capable LLMs that have already been instruction-tuned (Honovich et al., 2022a; Taori et al., 2023; Chiang et al., 2023; Peng et al., 2023). In contrast, GIFT4CODE generates synthetic data from vanilla LLMs that have not gone through instruction-tunning.

**Synthetic Data from LLMs**    Besides generating data for instruction following, a number of recent studies have also harnessed general-purpose LLMs to generate realistic synthetic data in areas where labeled data limited, such as language understanding and clinical research (Rosenbaum et al., 2022a; Tang et al., 2023; Borisov et al., 2022; Liu et al., 2022; Rosenbaum et al., 2022b; Josifoski et al., 2023). To improve the quality of synthetic data extracted from LLMs, such approaches usually apply a rejection sampling procedure and filter predictions based on domain-specific heuristics such as logical consistency (Bhagavatula et al., 2022; Yin et al., 2022c). GIFT4CODE is in spirit of this line in that it leverages program execution feedback to filter code predictions (Xu et al., 2020).

## 6    CONCLUSION

We have presented GIFT4CODE, a framework for instruction fine-tuning large language models of code in which the training is guided by execution based specifications. Empirically, we demonstrated how our approach enhances the quality of generated code, substantially improving accuracy on two challenging data science benchmarks, ARCADE and DS-1000.

**Limitations**    The Python types featured in ARCADE and DS-1000 still represent a limited subset of all potential Python types. Furthermore, I/O specifications may not always accurately capture a developer's true intent. We plan to extend our approach to consider a broader variety of Python types, including custom-defined classes, and to explore other types of specifications in future work.

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

## A    APPLYING GIFT4CODE TO DATA SCIENCE CODE GENERATION

In this appendix section, we demonstrate the practical application of our proposed approach GIFT4CODE, as discussed in §3, to a specific dataset, ARCADE (Yin et al., 2022b). We follow the same setup as described in §3. Our starting point involves a "Generalist LLM" and a Code LLM, the alignment of which we seek to improve.

### A.1    SYNTHETIC DATA GENERATION

We first gathered CSV files from diverse GitHub repositories focusing on data science projects. These CSV files encompass a wide range of data types, structures, and domains, serving as the programmatic context to initialize the synthetic data generation. Subsequently, we employed a "Generalist" LLM specifically designed for natural language understanding tasks, distinguished from the base code LLMs. This model was utilized to generate natural language questions based on the information extracted from the collected CSV files. For the following sections, we denote the set of natural number from 0 to $W$ and $\mathbb{N}_W = \{1, 2, 3, \ldots, W\}$.

**Creating NL Intents (Questions)**    Using the entire CSV file as the programmatic context to prompt the "Generalist" LLM is infeasible due to its length after tokenization. Instead, we extracted the header for each CSV file as the programmatic context $c_i$ to query the LLM to generate the NL intents $\{x_i\}$. Given the length of an entire CSV file after tokenization, using it as the programmatic context to prompt the "generalist" LLM is impractical. Instead, we derive the header and its first three rows from each CSV file as the programmatic context, denoted as $c_i$, to query the LLM to produce NL intents $x_i$. To construct the few-shot exemplars, we randomly selected a subset of CSV headers, denoted as $C' = \{c_i' \mid i \in \mathbb{N}_P\}$ where $P < 10$. The prime symbol $'$ in the superscript denotes the variable will be used in the few-shot exemplars. We manually crafted a set of 10 data science-related questions corresponding to each $c_i'$, denoted as $X_i' = \{x_i^j \mid j \in \mathbb{N}_{10}\}$. This process allowed us to create a few-shot prompt set $\mathcal{P}$ that consists of pairs of CSV headers and associated questions, formulated as $\mathcal{P} = \{(c_i', X_i')\}_{i=1}^P$.

We employed the standard few-shot prompting approach, which concatenates $\mathcal{P}$ to each CSV header $c_i$ when querying the LLM. In specific, each header $c_i$ is augmented to $\hat{c}_i = (\mathcal{P}, c_i)$. In this configuration, the LLM is prompted to generate questions related to $c_i$, while emulating the style of $\mathcal{P}$. After generation, we perform diversity filtering on $X_i$. Let $\widehat{X}_i$ denote the set of NL questions after the filtering process. We initialize $\widehat{X}_i$ as $\{q_0\}$ where $q_0 \sim \mathcal{U}(X_i)$), a question randomly selected from $X_i$. Similar to Wang et al. (2022a), we iterate over $\{X_i \setminus q_0\}$, a new question is added to the set $\widehat{X}_i$ only when its ROUGE-L overlap with any $q \in \widehat{X}_i$ is less than 0.7. Questions that are repetitive are also filtered out. For notation simplicity, in the following sections, we use $X_i$ to represent the set of NL intents after the filtering process. Empirically, we observe that the NL questions generated by this LLM encompassed a wide range of tasks relevant to the given programmatic context, as shown in Listing 2 and Listing 3.

**Predicting Code Solutions** Subsequently, we employed the code LLM, which we aim to improve, to generate code that addresses these NL questions. For each question $q_i^j \in X_i$ where $j$ denotes the $j^{\text{th}}$ question associated with the programmatic context $c_i$, we can draw code solutions from $P_{\text{LLM}}(y \mid c_i, q_i^j)$. This leads to a pool of code samples $\{y\}_i^j$. Following this, we applied heuristic filtering on these code samples, adopting two criteria: 1) Maximizing the diversity of Python library API calls, and 2) Ensuring the produced execution output is meaningful, for instance, yielding a `pandas.DataFrame` object if the code is executable. This process resulted in a synthetic dataset $\mathcal{D}$ that enables instruction fine-tuning. Notice that $\mathcal{D}$ does not contain any I/O specifications yet.

The synthetic data generation process leverages the capabilities of the LLM to create a diverse set of natural language intents that correspond to the header information extracted from the CSV files. By incorporating the knowledge and understanding of "Generalist" LLM, we are able to generate a rich and varied synthetic dataset that contains a wide range of data science tasks and scenarios.

**Executing Code Samples** Executing code derived from a pre-training corpus which was used to train the code LLM can be challenging, as it often demands notoriously complicated environment setup. In contrast, the synthetic code samples offer a significant advantage, as they can be executed more easily without the need for complex environment setup. Moreover, the execution of synthetic code samples enables us to track variables and their states both before and after running the code. This information can be utilized to identify input variables that exhibit state changes after execution. We label these variables as inputs to the code. In addition, by examining the final states of these input variables, we can effectively establish the input-output specifications of the code.

## B   LLM GENERATED NL INTENTS

In this section, we demonstrated the NL questions generated by the "generalist" LLM on the ARCADE dataset. To begin, we provide an exemplar used in the few-shot prompt when querying the LLM to generate NL questions, as seen in Listing 1. The given exemplar consists of the CSV header along with the first three rows. If any entry within the first three rows exceeds 50 characters in its string representation, we truncate it to maintain conciseness. As shown in Listing 1, we handcrafted 10 diverse questions, covering as many data science topics as possible.

In this section, we provide two examples of NL intent generation. Each of these examples, as shown in Listing 2 and Listing 3, includes both the programmatic context and the output generated by the LLM. Listing 2 illustrates an instance regarding a Pokémon game experience. Notably, the LLM tends to generate relatively basic questions in this context, which don't necessitate the use of complex Python APIs such as `pandas.groupby`. Conversely, Listing 3 presents an example related to a Netflix TV and movie dataset. In this case, the LLM produces more comprehensive questions. Addressing these questions requires multiple API calls, indicating a higher level of complexity.

## C   I/O SUMMARY EXAMPLES

To begin with, we showed an example of a few-shot exemplar used to prompt the "generalist" LLM in generating an I/O summary for the ARCADE dataset, as detailed in section 3.2. The exemplar in Listing 4 comprises an input dataframe schema, a Python code solution, execution output, and user intent. The anticipated response to this prompt is an I/O summary, outlining the input-output variable names and their types. In this example, there is only one variable — ""alc"" which is a `pandas.DataFrame`. Next, the LLM is expected to give a succinct description on the salient input columns, as well as a brief summary of the example output columns.

We present two examples of LLM generated I/O summaries on the ARCADE dataset, as illustrated in Listing 5 and Listing 6. As mentioned in §4.1, we deliberately obscure the execution output details when prompting the LLM to generate an I/O summary. This step helps to more realistically simulate user provided specifications. Each example consists of its prompt we used to query the LLM for the I/O summary, the resulting example augmented by the generated I/O summary and the ground truth execution output which was never exposed to LLM.

The first example Listing 5 focuses on a dataset detailing mobile phone information, with the user intent being to determine the quantity of different smartphones released each decade. The subsequent

```
1   First 3 rows from dataset 4-wheeler-EV.csv (column data types in parentheses)
2   | Comfort (float) | Performance (float) | Fuel Economy (float)
3   | Value for Money (float) | Condition (string) | driven (string) | rating (float)
4   | model_name (string) |
5   |----------------------------------------------------------------------------|
6   | 4 | 5 | 5 | 5 | New | Few hundred kilometers | 5 | hyundai kona |
7   | 1 | 1 | 1 | 1 | New | Haven't driven it | 0 | hyundai kona |
8   | 5 | 5 | 5 | 4 | New | Few thousand kilometers | 5 | hyundai kona |
9
10  First 3 rows from dataset 2-wheeler-EV.csv (column data types in parentheses)
11  | Owned for (string) | Ridden for (string) | rating (int)
12  | Visual Appeal (float) | Reliability (float) | Performance (float)
13  | Service Experience (float) | Extra Features (float) | Comfort (float)
14  | Maintenance cost (float) | Value for Money (float) | Model Name (string) |
15  |----------------------------------------------------------------------------|
16  | Never owned | nan | 1 | 3 | 4 | nan | nan | nan | 4 | nan | 1 | TVS iQube |
17  | > 1 yr | < 5000 kms | 1 | 3 | 1 | nan | 1 | nan | 3 | nan | 3 | TVS iQube |
18  | < 3 months | < 5000 kms | 3 | 4 | 4 | nan | 2 | nan | 5 | nan | 2 | TVS iQube |
19
20  Here are a series of contextually dependent data wrangling and exploratory data
21  analysis tasks for the dataset:
22  Task 1: How many new cars have a performance rating of more than 3?
23  Task 2: Among the Hyundai Kona reviews, do those with a purchase history have
24  average performance rating better than the ones that have not yet been purchased?
25  Task 3: Create a new column called "Review's Word Count" and Show the rating of
26  the shortest review
27  Task 4: Which model has the highest average word count?
28  Task 5: How many reviews are for used cars for this model?
29  Task 6: What are the top five models with most number of bikes having mileage
30  less than 5000 kilometers
31  Task 7: Which of these models has the highest comfort score on average?
32  Task 8: What are those models mostly used for?
33  Task 9: How many times the word scooter was mentioned in the bike's reviews?
34  Task 10: Show a list of words that were mentioned only once in the reviews
35  [END]
```

Listing 1: An exemplar in the few-shot prompt when querying LLM to generate NL intents.

```
1  | id (int) | identifier (string) | species_id (int) | height (int) | weight (int)
2  | base_experience (int) | order (int) | is_default (int) |
3  |-------------------------------------------------------------------------|
4  | 1 | bulbasaur | 1 | 7 | 69 | 64 | 1 | 1 |
5  | 2 | ivysaur | 2 | 10 | 130 | 142 | 2 | 1 |
6  | 3 | venusaur | 3 | 20 | 1000 | 263 | 3 | 1 |
```

(a) The programmatic context to query the "generalist" LLM, albeit only a subset of the columns are displayed.

```
1  Here are a series of contextually dependent data wrangling and exploratory data
2  analysis tasks for the dataset:
3  Task 1: How many pokemons are there in the Pokedex?
4  Task 2: Find the total number of columns that are integers.
5  Task 3: Calculate the ratio of mean weight to height for each pokemon
6  Task 4: What is the weight of "Snorlax" ?
7  Task 5: How many pokemon have the same average base experience as their id?
8  Task 6: Find the order for each pokemon that weighs less than 100 pounds
9  Task 7: What is the "mean" and "median" of "height" column ?
10 Task 8: Show the names of the pokemons with minimum and maximum weight, height
11 and base experience.
12 Task 9: Show the first 20 and last 10 pokemon with their average base experience.
13 Task 10: Create a new column called "size_cat" that has size categories for
14 pokemon (child: 1-10, teenager: 11-20, adult: 21+)
```

(b) LLM generated questions for the above programmatic context.

Listing 2: Programmatic context and generated questions on an example concerning pokemon game experience dataset.

```
1  | show_id (string) | type (string) | title (string) | director (string) |
2  | country (string) | date_added (string) | release_year (int) | ... |
3  |-------------------------------------------------------------------------|
4  | s1 | Movie | Dick Johnson Is Dead | Kirsten Johnson | United States |
5      September 25, 2021 | 2020 | PG-13 | 90 min | ... |
6  | s2 | TV Show | Blood & Water | nan | South Africa |
7      September 24, 2021 | 2021 | TV-MA | 2 Seasons | ... |
8  | s3 | TV Show | Ganglands | Julien Leclercq | nan |
9      September 24, 2021 | 2021 | TV-MA | 1 Season | ... |
```

(a) The programmatic context to query the "generalist" LLM, albeit only a subset of the columns are displayed.

```
1   Here are a series of contextually dependent data wrangling and exploratory data
2   analysis tasks for the dataset:
3   Task 1: What is the count of number of seasons for each show?
4   Task 2: How Many shows have "TV14" as rating?
5   Task 3: How many movies have a rating of 13+?
6   Task 4: Show the top 10 TV shows with most number of seasons
7   Task 5: Make a new column "genre" that combines all genres into one column.
8   Task 6: Show the names of casts who have been in at least 5 shows?
9   Task 7: How many TV Shows are there that have been released before 2017?
10  Task 8: For each director, how many shows have been added in 2020?
11  Task 9: Show the movies that have director's name with the letter 'b' in it.
12  Task 10: Show the number of shows released before 2020 in the genre "Documentaries"?
13  Task 11: For each type, what are the average durations?
14  (Show dataframe that has type and duration as columns)
```

(b) LLM generated questions for the above programmatic context.

Listing 3: Programmatic context and generated questions on an example concerning Netflix TV show and movie dataset.

I/O summary generated by the LLM identifies the "smartphone" and "year" columns as the most relevant from the input dataframe and describes the output values as being of int64 type, with an index year. The generated I/O summary correctly describes the ground truth presented in the bottom of Listing 5.

In the second example Listing 6, the LLM is dealing with movie data where the user's intent is to chronologically list the worldwide gross of films released each month from November 2021 to June 2022. The corresponding I/O summary generated by the LLM correctly identified the salient columns in the input dataframe, "Worldwide" and "Released Date". However, an observed discrepancy between the ground truth output and the I/O summary indicates that the LLM's generation is not entirely accurate for this specific example.

These two test examples on ARCADE illustrate the LLM's capabilities in generating I/O summaries, while highlighting the potential discrepancies that may occur.

## D    ADDITIONAL EXPERIMENTAL RESULTS

In this section, we offer two additional experiments that supplement the results presented in §4.2. We have demonstrated the GIFT4CODE model consistently outperforms in all types of I/O specifications. A natural follow-up question might be whether the instruction fine-tuning degrades the model's programming ability if no specifications are provided by the user. To address this concern, we have conducted further experiments where an instruction tuned model is evaluated on the ARCADE dataset, in the absence of any specification.

The results in Tab. 3 demonstrate that while the instruction tuned model does perform slightly worse than the model fine-tuned on data with no specifications, the difference is marginal. In the $pass@5$ and $pass@20$ settings, both with no context and full context, the model's performance (all types of specifications) only decreases by most 2% when compared with the model without specifications. This is expected as the discrepancy between the fine-tuning data and the testing data could lead to a minor regression in performance. This marginal decrease in performance is counterbalanced by the significant improvement we previously observed in §4.2. Therefore, GIFT4CODE with I/O summary still remains the superior method for instruction fine-tuning.

## E    BROADER IMPACTS

We outline the broader impacts of our research in both positive and negative aspects. On the positive side, our proposed approach, GIFT4CODE could significantly increase developers' productivity by enhancing the alignment of Large Language Models (LLMs) with user intentions in the context of code generation. It also lowers the barrier of entry to programming for non-experts. On the negative side, like any technology that automates human labor, tools which generate code might displace jobs. In our early investigation, this is still far from happening because the $pass@20$ accuracy on ARCADE is still less than 50% which is lower than an average Python expert can achieve.

A potential impact, which we regard as neutral, is the potential of our research to fundamentally reshape the nature of programming itself. By making the code generation more specification-driven, our approach aligns with the paradigm of declarative programming. It allows programmers to focus more on high-level problem-solving rather than the details of syntax. On the other hand, it still has certain degree of disconnection from the common coding style which is imperative. Therefore, the implications of this transformation are still unknown.

```
1   """
2   The input dataframe schema is:
3   Schema of Dataframes:
4   Columns in alc with example values:
5   country (Afghanistan), beer_servings (0), spirit_servings (0), wine_servings (0),
6   total_litres_of_pure_alcohol (0.0), continent (AS)
7   """
8
9   # The Python solution is:
10  import pandas as pd
11  alc = pd.read_csv("drinks.csv")
12  alc['continent'].unique()
13  alc.groupby('continent').agg({'beer_servings': np.mean}).sort_values(
14      'beer_servings', ascending=True)
15  alc.dropna()
16
17  # The execution output is:
18  alc:
19  |  beer_servings (float) |
20  |-----------------------|
21  |    102    |
22  |     20    |
23
24  # The user intent is:
25  Rank the continents that consume the least amount of beer on average.
26
27  # The I/O specification is:
28  alc: a pandas.core.DataFrame. Given the user intent and the code, the salient
29  columns (at most given 3) in the input dataframe are beer_servings, continent.
30  The output dataframe has columns (at most given 3) such as beer_servings.
```

Listing 4: An exemplar regarding `pandas.DataFrame`.

```
1   """
2   The input schema is:
3   # Schema of Dataframes:
4   # Columns in phones with example values:
5   # manufacturer (Nokia), model (1100), form (Bar), smartphone (No), year (2003)
6   """
7
8   # The Python solution is:
9   yearly_smartphones = phones.groupby(['year', 'smartphone'],
10      as_index=False).size().pivot_table(
11          index='year',columns='smartphone', values='size').fillna(0)
12  yearly_smartphones.groupby((yearly_smartphones.index//10)*10).Yes.sum()
13
14  # The execution output is:
15  __output__:
16  pandas.core.series.Series
17
18  # The user intent is:
19  How many different smartphones were released each decade?
20
21  The I/O specification is:
```

(a) The prompt used to query LLM for I/O summary.

```
1   # In[ ]:
2   import pandas as pd
3   import numpy as np
4   phones = pd.read_csv('dataset/best-selling-mobile-phones.csv')
5   # In[ ]:
6   # Schema of Dataframes:
7   # Columns in phones with example values:
8   # manufacturer (Nokia), model (1100), form (Bar), smartphone (No), year (2003)
9   # In[ ]:
10  phones[phones.form.str.lower().str.contains('touchscreen')].groupby(
11      'manufacturer').model.nunique().idxmax()
12  # In[ ]:
13  year_phones = phones[phones['year'] >= phones['year'].max()-15]
14  year_phones.groupby(['year','manufacturer','form'], as_index=False).size().pivot(
15      index=['year','manufacturer'], columns='form').fillna(0)
16
17  # In[ ]:
18  How many different smartphones were released each decade?
19  """
20  Input-output Summary:
21  __output__: a pandas.core.series.Series. Given the user intent and the code, the
22  salient columns (at most given 3) in the input dataframe are smartphone, year.
23  The output values are of type int64, with an index year. Here is my code solution:
24  """
25  # In[ ]:
```

(b) A test example on ARCADE augmented with a LLM generated I/O summary.

```
1   year
2   1990     0.0
3   2000    12.0
4   2010    58.0
5   Name: Yes, dtype: float64
```

(c) The ground truth of the output in the above test example.

Listing 5: An example of an I/O summary generated by the LLM on ARCADE for a pandas.Series.

```
1  """
2  The input schema is:
3  Schema of Dataframes:
4  Columns in df with example values:
5  Movie (JugJugg Jeeyo), Worldwide (74.5), India Net (50.24), India Gross (54.5),
6  Budget (100), Verdict (None), Movie Type (Bollywood), Released Date (24-Jun-22)
7  """
8
9  # The Python solution is:
10 df_t[['month', 'Worldwide']].groupby('month').sum().T
11
12 # The execution output is:
13 __output__:
14 pandas.core.frame.DataFrame
15
16 # The user intent is:
17 List the worldwide gross of the films released for each month since November,
18 2021 to June 2022. List  the months in chronological order.
19
20 # The I/O specification is:
```

(a) The prompt used to query LLM for I/O summary.

```
1  # In[ ]:
2  from datetime import datetime
3  import pandas as pd
4  df = pd.read_csv('dataset/bollywood_2022.csv')
5  # In[ ]:
6  # Schema of Dataframes:
7  # Columns in df with example values:
8  # Movie (JugJugg Jeeyo), Worldwide (74.5), India Net (50.24), India Gross (54.5),
9  # Overseas (20.0), Movie Type (Bollywood), Released Date (24-Jun-22)
10 # In[ ]:
11 df.columns = [column.replace(' ', '') for column in df.columns]
12
13 # In[ ]:
14 List the worldwide gross of the films released for each month # since
15 November, 2021 to June 2022. List the months in chronological order.
16 """
17 Input-output Summary:
18 __output__: a pandas.core.DataFrame. Given the user intent and the code,
19 the salient columns (at most given 3) in the input dataframe are Worldwide,
20 Released Date. The output dataframe has columns (at most given 3) such as month,
21 Worldwide. Here is my code solution:
22 """
23 # In[ ]:
```

(b) A test example on ARCADE augmented with a LLM generated I/O summary.

```
1  | April, 2022 (float) | December, 2021 (float) | February, 2022 (float)
2  | January, 2022 (float) | June, 2022 (float) | March, 2022 (float)
3  | May, 2022 (float) | November, 2021 (float) |
4  |-------------------------------------------------------------|
5  | 1748.13 | 10988.6 | 1812.99 | 114.12 | 4763.05 | 3849.25 | 7730.75 | 169.23 |
```

(c) The ground truth dataframe of the output in the above test example.

Listing 6: An noisy example of LLM generated I/O summary. The LLM generated I/O specification is inaccurate as evident from the discrepancies between the I/O summary in part (b) and the ground truth in part (c).

| Methods | ARCADE | | | |
| | pass@5 | | pass@20 | |
| | No Context | Full Context | No Context | Full Context |
| --- | --- | --- | --- | --- |
| *Synthetic Data Fine-tuning* | | | | |
| Code LLM (no spec.) | **20.78** | **34.33** | **33.40** | **46.94** |
| + TypeDesc | 20.59 | 33.43 | 32.98 | 46.01 |
| + I/O Examples | 20.04 | 31.76 | 31.40 | 43.60 |
| + I/O Summary | 20.04 | 32.39 | 32.24 | 45.14 |

Table 3: *pass@k* on ARCADE. For each type of I/O specification in Tab. 3 (*e.g.* +I/O Summary), intents are augmented with I/O specifications of that type (*e.g.* intents inline with I/O summary) in fine-tuning data. At test time, input intents do not contain have any specification.

